# The Effect of *Festulolium* Silage-Based Diets on the Content of Tocopherols, β-Carotene and Retinol in Meat from Young Rams

**DOI:** 10.3390/ani13111817

**Published:** 2023-05-31

**Authors:** Sylwia Czurgiel, Zofia Antoszkiewicz, Magdalena Mazur-Kuśnirek, Marek Bogdaszewski

**Affiliations:** 1Department of Animal Nutrition, Feed Science and Cattle Breeding, University of Warmia and Mazury in Olsztyn, 10-718 Olsztyn, Poland; 2Institute of Parasitology of the Polish Academy of Sciences, Research Station in Kosewo Górne, 11-700 Mrągowo, Poland

**Keywords:** lipophilic vitamins, provitamin A, biologically active compounds, muscle tissue

## Abstract

**Simple Summary:**

A *Festulolium* grass is an intergeneric hybrid of fescue (*Festuca* L.) and ryegrass (*Lolium* L.) that combines the nutritional qualities and weather resistance of each. Its high nutritional value and resistance to extreme weather conditions are important considerations in the era of global climate change. *Festulolium* has been used as a forage crop for many years, but little information is available on the levels of biologically active compounds (β-carotene and tocopherols) in *Festulolium*-based diets and in the meat of animals fed such diets. The present study assessed the contents of β-carotene, retinol and tocopherols and the proximate chemical composition of meat from young rams receiving diets based on silage made from different *Festulolium* cultivars (Becva, Felopa, Paulita), as compared with meat from rams fed a diet based on hybrid ryegrass. The meat of animals fed *Festulolium* silage-based diets contained more tocopherols and retinol than the meat of sheep receiving a hybrid ryegrass-based diet. Meat from rams fed diets based on silage made from *Festulolium* cvs. Felopa and Paulita was characterized by the highest concentrations of tocopherols and retinol. The results of this study indicate that *Festulolium* silage can be a desirable component of ram diets, but its efficacy may vary across cultivars.

**Abstract:**

This study analyzed the chemical composition and the β-carotene, tocopherol and retinol content of meat (*musculus longissimus lumborum*, *musculus gluteus maximus*) from young rams fed different silage-based diets. The animals were divided into four groups depending on the type of silage in the feed ration (hybrid ryegrass, *Festulolium* cv. Becva, *Festulolium* cv. Felopa, *Festulolium* cv. Paulita). The concentrations of the examined lipophilic compounds were measured via high-performance liquid chromatography (HPLC). Animals fed a diet based on ryegrass silage had the highest intakes of α-, β-, γ- and δ-tocopherol and total tocopherols (*p* < 0.01). Muscles of rams fed *Festulolium* silage-based diets contained more tocopherols (*p* < 0.01) and retinol (*p* < 0.01 in *m. g. maximus*; *p* = 0.053 in *m. l. lumborum*) than the meat of animals receiving hybrid ryegrass-based diets. The highest amounts of tocopherols (*p* < 0.01) and retinol (*p* < 0.01 in *m. g. maximus*; *p* = 0.053 in *m. l. lumborum*) were found in meat from rams fed rations based on silage made from *Festulolium* cvs. Felopa and Paulita. The analysis revealed a positive correlation between the intake of tocopherols and the α-tocopherol content of meat (r = 0.99), as well as between the β-carotene intake and the retinol content of meat (r = 0.92 in *m. l. lumborum*; r = 0.91 in *m. g. maximus*).

## 1. Introduction

Grasses of the genus *Festulolium* are intergeneric hybrids of fescue (*Festuca L.*) and ryegrass (*Lolium L.*) [1]. Commercial *Festulolium* cultivars are widely grown in Scandinavia (Denmark, Norway, Sweden, Finland) and in other European countries (United Kingdom, Germany, Czechia, Poland) [2,3]. The interest in *Festulolium* has increased considerably in recent years. Various potential uses of *Festulolium* have been tested, including energy generation, grassland restoration and reducing water runoff [4]. *Lolium* x *Festuca* hybrids have been used in feed production for several decades. They combine the complementary traits of ryegrass and fescue, including high nutritional value and resistance to extreme weather conditions [3]. *Festulolium* hybrids respond well to drought, frost and periodic excess soil moisture [5,6,7,8]. They are characterized by high productivity and persistence in the sward [9,10,11,12,13,14]; they are also a source of valuable fodder for ruminants, with high concentrations of crude protein and soluble carbohydrates and a low concentration of crude fiber [3]. Plant adaptation to environmental stresses is an important consideration in the era of global climate change [15]. According to the literature, extreme weather events have already contributed to a decline in the productivity of permanent grasslands [16,17,18]. Progressive global warming poses a threat to many plant species in Europe [19] and may affect crop quality [20]. 

*Festulolium* can be used in the production of both green fodder and preserved feed, alone or in mixtures with red clover and alfalfa [3]. *Festulolium* roughage can be fed fresh or ensiled, and it can be offered to cattle and sheep as hay and haylage [3,9]. In Poland, fattening lambs are usually fed hay or haylage complemented with oat or barley grain [21]. Roughage is cheap and readily available and constitutes a rich source of nutrients and bioactive compounds, including carotenoids and tocopherols [22]. Tocopherols (vitamin E), as well as vitamin A (and its precursor, β-carotene), are essential nutrients with important functions in many biological processes. Both components are lipophilic and share several mechanisms underlying their metabolism. Vitamins A and E are potent antioxidants that boost immunity, protect body tissues against oxidative stress and are essential for normal reproduction, growth and development [23]. Both compounds (vitamins A and E) must be provided in the diet since they cannot be synthesized by the body [22,23]. It has been reported that vitamins A and E accumulate in the tissues and are transferred to animal products, such as meat, and are known to improve its shelf life and nutritional value, affect its color, odor, flavor and texture and delay rancidification [22]. Due to the above reasons, *Festulolium* can be a source of fodder for ruminants, including sheep. *Festulolium*-based rations have been thoroughly investigated, but only a few studies have focused on the content of bioactive compounds, including carotenoids and tocopherols in *Festulolium* diets. The information on the effect of *Festulolium*-based diets on the content of bioactive compounds in ram meat is also scant.

The aim of this study was to evaluate the influence of *Festulolium* silage-based diets on the content of basic nutrients, β-carotene, tocopherols and retinol, in the *musculus longissimus lumborum* (MLL) and *musculus gluteus maximus* (MGM) of young rams. Hybrid ryegrass and three *Festulolium* cultivars (Becva, Felopa, Paulita), most commonly grown in Central Europe, were used in the study.

## 2. Materials and Methods

### 2.1. Experimental Silages

Four silages used in the experiment were produced in the Experimental Station in Tomaszkowo (53°43′ N, 20°24′ E; University of Warmia and Mazury, Olsztyn, Poland). The following grasses were used:Third-harvest herbage of hybrid ryegrass cv. Bakus (group RB);Third-harvest herbage of three *Festulolium* cultivars:Becva (*Festuca arudinacea* × *Lolium multiflorum*) (group FB);Felopa (*Festuca pratensis* × *Lolium multiflorum*) (group FF);Paulita (*Festuca pratensis* × *Lolium multiflorum*) (group FP).Seeds of hybrid ryegrass and three *Festulolium* cultivars were sown in experimental plots with an area of 544 m^2^ (16 m × 34 m; seeding rate-approx. 4.65 g seeds/m^2^) on 11 April 2015. Before sowing, all plots were fertilized with N, P and K (at 2.0, 4.0 and 3.6 g/m^2^, respectively). In the first growing season, grass was cut at the end of August 2016, and left in the field. After 24 h of wilting, herbage was harvested with a round baler (Kverneland Group, Klepp, Norway). Six layers of plastic film were used to wrap the bales (30 μm × 750 mm) with a stationary bale wrapper (Sipma S.A., Lublin, Poland). The time between baling and wrapping did not exceed 60 min.

### 2.2. Animals, Experimental Design and Diets

The experiment was carried out on 24 young Berrichon du Cher rams, in the experimental station of the Department of Animal Nutrition, Feed Science and Cattle Breeding (University of Warmia and Mazury in Olsztyn). During the study, the animals were cared for in accordance with the Act of 15 January 2015 on the Protection of Animals Used for Scientific or Educational Purposes [24]. The rams were weaned at around 100 days of age. At the beginning of the experiment, the average body weight of animals in each group was as follows: RB, 33 ± 6 kg; FB, 32 ± 5 kg; FF, 31 ± 6 kg; FP, 32 ± 4 kg. The lambs were kept in individual cages (one animal in each cage, 2 × 3 m in size, made of stainless steel) for 50 days (including a 10-day adaptation period). The animals had free access to feed and water (via automatic drinkers). Each animal was given feed in a trough placed in the cage. Before putting the feed into each trough, each feed was weighed. The uneaten feed from each trough was weighed each day before new feed was given. The feed was distributed daily, twice a day (in the morning at 7 a.m. and in the afternoon at 5 p.m.), in sufficient quantity to leave some uneaten feed. Feed intake (calculated as the difference between the weight of feed given to each animal and the weight of uneaten feed) and the amount of uneaten feed were recorded throughout the study. The chemical composition of the rations was formulated according to the DLG system, assuming a daily gain of 300 g [25]. The lambs were randomly divided into four equal groups (six animals per group). Rations containing different silages were used in the experimental groups: hybrid ryegrass cv. Bakus, *Festulolium* cv. Becva, *Festulolium* cv. Felopa, *Festulolium* cv. Paulita (groups RB, FB, FF and FP, respectively; Table 1; Figure 1). The diets also contained ground barley grain, soybean meal and a mineral–vitamin premix (administered manually at 20 g/animal). The roughage-to-concentrate ratio was 60:40 on a dry matter (DM) basis. After 50 days of the experiment, all rams (*n* = 24) were sacrificed in the Cattle Slaughterhouse in Biskupiec (*Zakład Uboju Bydła Biskupiec*, *Sp. z o. o*.; 0°58′ E, 53°52′ N), in accordance with Council Regulation (EC) No. 1099/2009 of 24 September 2009 on the protection of animals at the time of slaughter [26] and Annex IV “Methods of Killing Animals” to the Directive 2010/63/EU of the European Parliament and of the Council of 22 September 2010 on the Protection of Animals Used for Scientific Purposes [27]. The carcasses were divided into half-carcasses that were chilled for 24 h (+4 °C). Samples of MLL and MGM were collected from the half-carcasses.

### 2.3. Feed Sampling and Analyses

Feed samples were dried at a temperature of 60 °C in Binder dryers (Binder GmbH, Tuttlingen, Germany), and they were ground in a cutting mill (Retsch, Haan, Germany) to a 1 mm particle size. The proximate chemical composition of dried and ground samples was determined through standard methods (Table 2) [28]. The concentration of neutral detergent fiber (NDF) was determined using the method proposed by Van Soest et al. [29] with the ANKOM 220 fiber analyzer (ANKOM Technology Corp., Macedon, NY, USA). The content of β-carotene and tocopherols was determined in fresh feed samples (immediately after collection) under limited exposure to sunlight. Feed samples were extracted with a mixture of petroleum ether and ethanol (96%) (1:1 *v*/*v*) at room temperature, in the dark, for 18 h. The obtained extracts were saponified with ethanolic 10% KOH solution for 2 h in the dark, under nitrogen atmosphere, using a vortex (IKA Poland Sp. z o. o., Warsaw, Poland), and extracted with ethyl ether and then repeatedly with petroleum ether. The prepared extracts were rinsed with 10% aqueous NaCl solution and repeatedly with deionized water. The eluates were dehydrated with anhydrous sodium sulfate and evaporated to dryness at 40 °C on a rotary evaporator (Janke & Kunkel IKA- Labortechnik, Staufen, Germany). The residue was dissolved in 5 cm^3^ hexane. The tocopherol content of feed samples was determined via reversed-phase high-performance liquid chromatography (RP-HPLC; Shimadzu, Kyoto, Japan). A Nucleosil C18 column was used, and the mobile phase was methanol:H_2_O (95: 5 *v*/*v*), at a flow rate of 1 cm^3^ min^−1^. An RF Ex 293 and Em 326 tocopherol detector and external standards were used: (±)-α-tocopherol (DL-all-rac α-tocopherol), β-tocopherol, (+)-γ-tocopherol and (+)-δ-tocopherol (SIGMA-ALDRICH, St. Louis, MO, USA) [30]. For the determination of β-carotene concentration, hexane-extracted feed samples were separated via HPLC, using a Gemini 5 μm column (Phenomenex, California, USA), C18, 110Å, 250 × 4 mm; the mobile phase was acetonitrile:methanol:dichloromethane (750:200:50 *v*/*v*/*v*; HPLC SIGMA-ALDRICH, St. Louis, MO, USA) at a flow rate of 1 cm^3^ min^−1^, UV-vis 450 nm detector, 20 μL loop, external standard-β-carotene type I, synthetic (SIGMA-ALDRICH, St. Louis, MO, USA) [31].

### 2.4. Meat Sampling and Analyses

Samples of MLL and MGM were collected after 24 h carcass chilling at 4 °C, from the right side of each carcass (at the last rib). Muscle tissue samples were packaged in polyethylene bags, transported to the laboratory in isothermal containers on ice and frozen until analysis (approx. seven days). The proximate chemical composition of meat was determined using standard methods [28]. The levels of β-carotene, retinol and tocopherols were determined under limited exposure to sunlight in meat samples that had been thawed before analysis and ground using a blender (Robert Bosch GmbH, Gerlingen, Germany). The obtained homogenates of MLL and MGM samples (Ultra-Turrax T25 homogenizer, Janke & Kundel IKA- Labortechnik, Staufen, Germany) with a mass of 1 g were saponified with 2 cm^3^ of a saponification mixture (375 g KOH, 750 cm^3^ H_2_O, 450 cm^3^ methanol) in a shaking water bath (200 rpm, 80 °C, 15′; Julabo GmbH, Seelbach, Germany). The samples were then cooled, 5 cm^3^ of 35% ethanol and 5 cm^3^ of 10% NaCl were added and the samples were shaken (vortex-IKA Poland Sp. z o. o., Warsaw, Poland) and left for 5′. Four cubic centimeters of n-hexane was added to the samples, which were shaken again and centrifuged at 1500× *g*, for 15′ (MPW Med. Instruments, Warsaw, Poland). Next, 3 cm^3^ of the extract was collected from the resulting supernatant, and extraction with 4 cm^3^ of n-hexane was repeated. The extracts were evaporated to dryness under a stream of N_2_. The residue was dissolved in 1 cm^3^ of n-hexane and filtered through a syringe filter [32,33,34]. The content of tocopherols and retinol in hexane-extracted samples was determined via RP-HPLC (Shimadzu, Kyoto, Japan). A Nucleosil C18 column was used, with a mobile phase of methanol:H_2_O (95: 5 *v*/*v*) and a flow rate of 1 cm^3^ min^−1^. For the RF Ex 293 and Em326 tocopherol detector and external standards, (±)-a-tocopherol (DL-all-rac a-tocopherol), b-tocopherol, (+)-g-tocopherol and (+)-d-tocopherol were used [30]. Retinol was determined with UV-VIS 336 nm detection and a synthetic retinol external standard (SIGMA-ALDRICH, St. Louis, MO, USA) [35]. The β-carotene content of the samples was determined via HPLC (Shimadzu, Kyoto, Japan). A Phenomenex Gemini 5 μm, C18, 11, 250 × 4 mm column with an acetone-trile:methanol:dichloromethane mobile phase (750:200:50 *v*/*v*/*v*; HPLC grade; SIGMA-ALDRICH, St. Louis, MO, USA) was used. The column flow rate was 1 cm^3^ min^−1^. A 450 nm UV-vis detector was used, with a 20 μL loop and a synthetic, external β-carotene type I standard (SIGMA-ALDRICH, St. Louis, MO, USA) [31].

### 2.5. Statistical Analysis

The results were processed statistically through one-way analysis of variance (ANOVA), linear correlation analysis and Duncan’s and Shapiro-Wilk test (*p* > 0.05–normal distribution). Homogeneity of variance was verified using Laven’s test. Arithmetic means (x) and the standard error of the mean (SEM) were calculated at a significance level of *p* ≤ 0.01. All calculations were performed using STATISTICA version 13.3 software.

## 3. Results

The highest intakes of total tocopherols, α-tocopherol and β-tocopherol were noted in rams fed a ration containing hybrid ryegrass silage (*p* < 0.01, Table 3). The intake of the above compounds was similar in the FP group (ration based on silage made from *Festulolium* cv. Felopa). Rams in the FP group consumed the largest amounts of γ- and δ-tocopherol, compared with groups RB, FB and FF (*p* < 0.01). The intake of the examined compounds by animals in groups FB and FF was similar and much lower than that in groups RB and FP. Differences in β-carotene intake between groups were not significant.

The proximate chemical composition of MGM samples collected in groups RB, FB and FF was similar (Table 4). The meat of rams fed silage made from *Festulolium* cv. Paulita (group FP) contained less fat and DM than other MGM samples, but the noted differences were not significant. An analysis of MGM samples revealed significant differences in the content of α-, β-, γ- and δ- tocopherol, total tocopherols and retinol (*p* < 0.001). The samples collected from rams fed silage made from *Festulolium* cv. Felopa (FF group) had the highest tocopherol content. The concentration of vitamin E was only slightly lower in MGM samples collected from group-FP animals (silage made from *Festulolium* cv. Paulita). The concentration of α-tocopherol in MGM samples was higher in the FF group than in the FP, FB and RB groups by 0.27, 0.74 and 0.92 mg/kg, respectively. The concentrations of γ- and δ-tocopherol were somewhat higher in the FF group than in the remaining groups. Retinol content was many-fold higher in MGM samples collected from group-FP rams (silage made from *Festulolium* cv. Paulita) than in those collected in the FB and RB groups (2.26 mg/kg vs. 1.11 and 0.77 mg/kg, respectively). Retinol concentration in MGM was similar in the FF and FP groups (2.12 and 2.26 mg/kg, respectively). None of the analyzed MGM samples contained β-carotene.

The proximate chemical composition of MLL samples collected from rams varied across groups, but no significant differences were observed (Table 5). The protein content of meat was slightly lower in the FF group (silage made from *Festulolium* cv. Felopa), and the fat content of meat was lower in the FP group (silage made from *Festulolium* cv. Paulita), relative to the other groups. It should be noted that total tocopherol concentrations in MLL samples were twice as high in the FF and FP groups than in the RB and FB groups (*p* < 0.001). The content of α-tocopherol in MLL samples was also significantly higher (*p* < 0.001), and the concentrations of β-, γ- and δ-tocopherol were several-fold higher in the FF and FP groups. The retinol content of MLL was higher in rams fed silage made from *Festulolium* cv. Paulita (group FP) than in those receiving silage made from hybrid ryegrass (group RB) or *Festulolium* cv. Felopa (group FF) (by approx. 64% and 55%, respectively), but the noted differences were not significant. None of the analyzed MLL samples contained β-carotene.

The α-tocopherol content of MGM samples was very strongly positively correlated with total tocopherol intake (Table 6). Very strong positive correlations were found between the α-tocopherol intake vs. the concentrations of β-tocopherol and α- and δ-tocopherol in MGM (r = 0.94 and r = 0.90, respectively). Very strong positive correlations were also observed between α-tocopherol and retinol (r = 0.93), between total tocopherols and retinol (r = 0.92) and between β-carotene intake and retinol in meat. Very strong negative and significant correlations were noted between β-tocopherol and γ-tocopherol and between β-tocopherol and total tocopherols. Fairly strong positive correlations were found between total tocopherols vs. γ-tocopherol and δ-tocopherol (r = 0.88 and r = 0.89, respectively; *p* < 0.05).

Vitamin concentrations in nutrient intake and MLL samples were significantly correlated (Table 7). The strongest correlations were observed between the α-tocopherol intake and total tocopherol concentrations in MLL, between α-tocopherol and δ-tocopherol and between α- and β-tocopherol vs. γ-tocopherol (all correlations were very strong and positive). A very strong correlation was noted between β-carotene intake and retinol concentration in MLL. Total tocopherol concentrations in MLL were very strongly positively correlated with the intake of β- and δ-tocopherol (*p* < 0.05). A fairly strong positive correlation was found between the β-tocopherol intake and the α-tocopherol concentration in meat (r = −0.86, *p* < 0.05). A moderate positive correlation was observed between the β-tocopherol concentration in meat and the δ-tocopherol intake (r = 0.7, *p* < 0.05). Weak, non-significant correlations were noted between retinol vs. γ- and δ-tocopherol. Retinol concentration was not correlated with the remaining tocopherol forms.

## 4. Discussion

### 4.1. Tocopherol Concentrations in Meat from Rams (Musculus Gluteus Maximus and Musculus Longissimus Lumborum) Fed Festulolium Silage-Based Diets

Vitamin E (in particular α-tocopherol) is a powerful antioxidant that neutralizes peroxyl free radicals responsible for damage to cellular structures, DNA and lipid oxidation [23]. The oxidation of meat lipids results in a number of compounds responsible for the formation of characteristics unacceptable to consumers, including a rancid, undesirable odor and taste [21,22]. The oxidation of meat lipids also negatively affects its color, texture and nutritional value, as unsaturated fatty acids and vitamins are degraded. Research into the possibility of extending the shelf life of meat products indicates that fat should be protected against oxidation [22]. MGM samples collected from animals receiving *Festulolium* silage-based diets were characterized by higher concentrations of total tocopherols and α-tocopherol than MGM samples collected from rams fed silage made from hybrid ryegrass (group RB). Optimal results were obtained when silage was made from *Festulolium* cvs. Felopa and Paulita—the concentrations of total tocopherols and α-tocopherol were higher by around 53% and 38%, as well as 47% and 67%, respectively, compared with those noted in MGM samples in the RB group. The concentration of vitamin E, including α-, β-, γ- and δ-tocopherol, in ruminant meat remains insufficiently investigated. Most research has focused on α-tocopherol, which predominates in animal tissues; other studies have also discussed γ-tocopherol [36]. For instance, the MLL of sheep fed silage made from *Festuca rubra,* alfalfa and red clover contained 6.06, 2.75 and 3.83 mg/kg α-tocopherol, respectively [21]. Luciano et al. [37] reported that the *longissimus thoracis et lumborum* of young rams fed different silage mixtures (timothy grass, timothy grass + sainfoin, timothy grass + red clover, timothy grass + sainfoin + red clover, sainfoin + red clover) contained 0.43–3.22 mg/kg of α-tocopherol and 0.01–0.03 mg/kg of γ-tocopherol. In the present study, the correlation analysis revealed a very strong positive correlation between the α-tocopherol content of meat and total tocopherol intake (Table 5), which corroborates the findings of Yang et al. [38]. Previous research has demonstrated that in ruminants, the α-tocopherol content of meat is directly correlated with the dietary intake of α-tocopherol [39,40,41]. Such a relationship was not observed in this study. In the RB, FB, FF and FP groups, the average daily intake of total tocopherols reached approximately 10, 7.1, 7.5 and 9.6 mg, respectively, and the average daily intake of α-tocopherol was determined to be 6.8, 4.8, 5.5 and 6.2 mg, respectively (Table 3, *p* < 0.01). In turn, in the RB, FB, FF and FP groups, total tocopherol concentrations in the MGM of rams reached 1.92, 2.09, 2.49 and 2.65 mg/kg, respectively, and the respective values for α-tocopherol were 1.38, 1.56, 2.30 and 2.03 mg/kg (Table 5). The meat of rams that consumed the greatest quantities of total tocopherols and α-tocopherol (group RB) contained the lowest concentrations of these compounds. The inclusion of *Festulolium*-based silage in ram diets increased the concentrations of both total tocopherols and α-tocopherol in meat, despite their lower intake. The highest total tocopherol concentrations were noted in the MGM of rams fed silage made from *Festulolium* cv. Paulita (group FP), which provided the greatest quantities of these compounds. However, the α-tocopherol content of meat samples collected in the FP group was not the highest despite its highest dietary intake in this group (*p* < 0.01). This phenomenon has not been elucidated in the literature to date. Red clover silage induced a decrease in the α-tocopherol concentration in meat from young rams [37] and in bovine milk [42]. According to Luciano et al. [37], polyphenol oxidase and/or isoflavones present in some silages may disrupt the intestinal absorption of α-tocopherol and its metabolism, thus decreasing its accumulation in tissues.

There is little information available on the concentrations of β-, γ- and δ-tocopherol in meat from rams fed diets containing silage. In a study by Valenti et al. [43], the content of γ- and δ-tocopherol in the meat of young rams was 1.99–3.66 ng/g and 19.9–28.68 ng/g (β form was not analyzed), but the animals received concentrate-based diets. It should be noted that most of the ingested β-, γ- and δ-tocopherol is secreted into bile or is not taken up and is excreted in the feces [44]. In the current experiment, higher dietary intake of β-, γ- and δ-tocopherol was not accompanied by the increased accumulation of these compounds in muscle tissue. For example, the average daily intake of β-tocopherol in the RB, FB, FF and FP groups was 0.87, 0.55, 0.58 and 0.63 mg/kg DM, respectively, whereas its concentration in MGM was 0.1, 0.1, 0.07 and 0.09 mg/kg, respectively (Table 4).

The proximate chemical composition of the MLL of rams did not vary significantly across groups (Table 4). The concentrations of total tocopherols, α-tocopherol and other tocopherol forms were considerably higher in MLL and in MGM samples. The MLL of rams fed silage made from hybrid ryegrass (group RB) was characterized by the lowest accumulation of total tocopherols and α-tocopherol, despite their highest dietary intake. The concentrations of total tocopherols and α-tocopherol were highest in the MLL of rams in the FF group (silage made from *Festulolium* cv. Felopa), where the dietary intake of these compounds was not highest. The dietary intake of both total tocopherols and α-tocopherol was highest in the FP group, but their concentrations in MLL samples were slightly lower than in animals fed an FF diet.

### 4.2. Retinol Concentration in Meat from Rams (Musculus Gluteus Maximus and Musculus Longissimus Lumborum) Fed Festulolium Silage-Based Diets

The metabolism and accumulation of β-carotene and retinol in animal tissues have not been thoroughly analyzed to date, and they vary considerably across animal species [45,46,47]. For instance, the predominant carotenoids in adipose tissue and blood serum are β-carotene in cattle and lutein in sheep and goats [46,48,49]. β-carotene is present in trace amounts in sheep tissues. Upon absorption, it is almost entirely converted to retinol [47,50]. In the current study, β-carotene was not detected in any of the analyzed meat samples (Table 4 and Table 5), which is consistent with the results of previous studies [46,47]. Retinol deposition in animal tissues is affected mostly by the diet, including β-carotene (provitamin A) levels, supplementation levels and an interaction between carotenoids and tocopherols [38,47,51]. Vitamin A has antioxidant properties and delays oxidative processes, thus affecting the sensory characteristics and chemical composition of meat and, consequently, its nutritional value [47]. The correlation analysis revealed a very strong positive correlation between β-carotene intake and retinol concentration in the meat of rams (Table 5 and Table 6). The results of few studies investigating the effect on feeding silage to sheep on retinol accumulation in muscle tissue are inconclusive. In the work of Luciano et al. [37], the retinol concentration in the intramuscular fat of lambs fed different silage mixtures ranged from 0.21 to 0.39 μg/g, but the noted differences were not significant. Kilinç et al. [52] demonstrated that the retinol concentration in the meat of yearling sheep fed diets based on alfalfa silage ranged from 2.91 to 8.33 µg/100 g depending on the type of tissue. In the present study, *Festulolium* silage-based diets affected the retinol concentration only in MGM (Table 4 and Table 5). It should also be stressed that higher dietary intake of β-carotene by rams was accompanied by a higher retinol concentration in MGM, but only in groups receiving *Festulolium* silage-based diets. Retinol concentration was highest in MGM samples collected from animals fed an FP diet (silage made from *Festulolium* cv. Paulita), which supplied the greatest amount of β-carotene (approx. 330 mg/day). The retinol content of MGM in the FF group (silage made from *Festulolium* cv. Felopa) was similar to that noted in the FP group (2.12 mg/kg and 2.26 mg/kg, respectively), although dietary β-carotene intake was lower in the former group by around 56 mg/day. Retinol concentration was lowest in the MGM of rams fed silage made from hybrid ryegrass (group RB), whereas β-carotene intake was higher in this group than in the FF group (322 mg/day vs. 274 mg/day). Previous research has shown that for some reasons, silage carotenes may be very poorly utilized or silage rations may increase the expenditure of vitamin A [53]. These observations should be confirmed in future feeding trials involving hybrid ryegrass silage.

## 5. Conclusions

The meat of young rams fed *Festulolium* silage-based diets contains more tocopherols and retinol than the muscles of animals receiving a hybrid ryegrass-based diet. Ram meat is highest in tocopherols and retinol when silage is made from *Festulolium* cvs. Felopa and Paulita. The results of feeding silage made from *Festulolium* cv. Becva and hybrid ryegrass are comparable (the former contributes to a slightly higher content of tocopherols and retinol in meat). The present findings indicate that *Festulolium* silage can be a valuable component of ram diets, which may contribute to the popularization of *Festulolium* cultivation both in Poland and worldwide. The results of this study suggest that the quality of meat from sheep fed rations containing *Festulolium* silage can be more stable, and its attributes can be preserved over longer storage periods, which are important considerations for both the processing industry and consumers.

## Figures and Tables

**Figure 1 animals-13-01817-f001:**
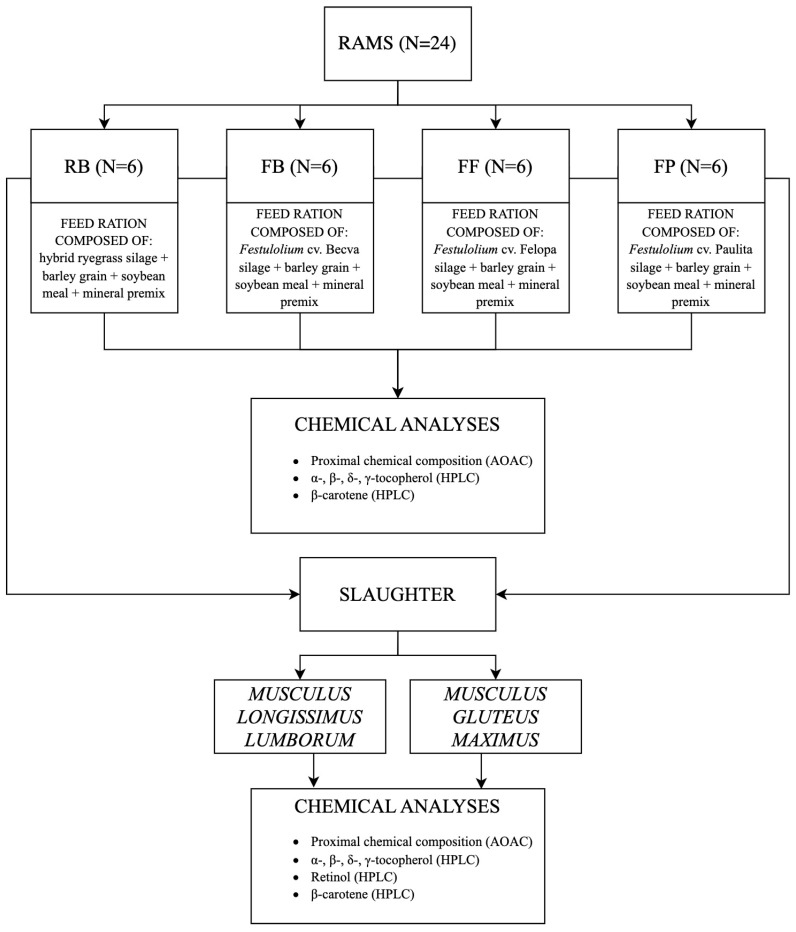
Simplified experimental design.

**Table 1 animals-13-01817-t001:** Composition of experimental diets fed to rams (% DM).

Specification	Groups			
	RB	FB	FF	FP
Diet composition, % DM				
Silage made from hybrid ryegrass cv. Bakus	60	-	-	-
Silage made from *Festulolium* cv. Becva	-	60	-	-
Silage made from *Festulolium* cv. Felopa	-	-	60	-
Silage made from *Festulolium* cv. Paulita	-	-	-	60
Barley grain	35	35	35	35
Soybean meal	5	5	5	5
Minerals and vitamins (g/d/animal)	20	20	20	20

Groups: RB, hybrid ryegrass cv. Bakus; FB, *Festulolium* cv. Becva; FF, *Festulolium* cv. Felopa; FP, *Festulolium* cv. Paulita. Premix composition (g/kg): vitamin E, 0.1; Fe, 2.5; Zn, 1; Mn, 1.5; Cu, 0.2; I, 0.05; Co, 0.04; Se, 0.02; Ca, 280; P, 70; Na, 40; Mg, 10.

**Table 2 animals-13-01817-t002:** Proximate chemical composition (g kg^−1^ DM) and the content of tocopherols and β-carotene in different silage-based diets fed to rams (mg kg^−1^ DM).

Specification	Groups				SEM
	RB	FB	FF	FP	
DM, g kg^−1^	429	435	430	431	0.342
OM	905	932	938	936	0.054
CP	149	148	130	134	0.202
EE	24	27	29	26	0.260
NDF	395	394	445	409	0.352
α-Tocopherol	5.89	4.44	5.10	5.52	0.220
β-Tocopherol	0.75	0.51	0.54	0.56	0.035
γ-Tocopherol	1.16	1.12	0.79	1.37	0.085
δ-Tocopherol	0.89	0.55	0.52	1.12	0.091
Total tocopherols	8.69	6.62	6.95	8.57	0.346
β-Carotene	278.39	241.84	253.68	294.34	0.497

Groups: RB, hybrid ryegrass cv. Bakus; FB, *Festulolium* cv. Becva; FF, *Festulolium* cv. Felopa; FP, *Festulolium* cv. Paulita. DM, dry matter; OM, organic matter; CP, crude protein; EE, ether extract; NDF, neutral detergent fiber. SEM—standard error of the mean.

**Table 3 animals-13-01817-t003:** Average daily intake of DM (g/day), β-carotene and tocopherols (mg/day).

Specification	Groups				SEM	*p*-Value
	RB	FB	FF	FP		
DM	1155	1075	1080	1120	3.617	0.104
CP	172	159	140	150	4.767	0.119
EE	28	29	31	29	1.372	0.920
α-Tocopherol	6.8 ^A^	4.8 ^B^	5.5 ^B^	6.2 ^AB^	0.269	<0.001
β-Tocopherol	0.87 ^A^	0.55 ^B^	0.58 ^B^	0.63 ^B^	0.040	<0.001
γ-Tocopherol	1.34 ^ABC^	1.2 ^B^	0.85 ^D^	1.54 ^A^	0.087	<0.001
δ-Tocopherol	1.03 ^A^	0.59 ^B^	0.56 ^B^	1.25 ^A^	0.094	<0.001
Total tocopherols	10.04 ^A^	7.12 ^B^	7.51 ^B^	9.6 ^A^	0.431	0.002
β-Carotene	322	260	274	330	4.409	0.239

Different superscripts within the same row are significant or trending (A/BC/D: *p* ≤ 0.01). SEM—standard error of the mean. Groups: RB, hybrid ryegrass cv. Bakus; FB, *Festulolium* cv. Becva; FF, *Festulolium* cv. Felopa; FP, *Festulolium* cv. Paulita. DM, dry matter; CP, crude protein; EE, ether extract. Nutrient intake was calculated based on feed intake and nutrient concentration in the ration.

**Table 4 animals-13-01817-t004:** Proximate chemical composition (%) and the content of β-carotene, tocopherols and retinol (mg/kg) in the *musculus gluteus maximus* of rams.

Specification	Groups				SEM	*p*-Value
	RB	FB	FF	FP		
DM	26.42	26.82	26.41	24.53	0.373	0.054
CA	1.11	1.14	1.08	1.03	0.014	0.060
CP	20.08	20.71	19.87	20.03	0.179	0.388
EE	5.23	4.97	5.46	3.47	0.412	0.330
α-Tocopherol	1.38 ^B^	1.56 ^B^	2.30 ^A^	2.03 ^A^	0.090	<0.001
β-Tocopherol	0.10 ^A^	0.10 ^A^	0.07 ^B^	0.09 ^A^	0.002	<0.001
γ-Tocopherol	0.27 ^B^	0.24 ^B^	0.34 ^A^	0.32 ^A^	0.012	<0.001
δ-Tocopherol	0.17 ^B^	0.20	0.23 ^A^	0.21	0.006	0.003
Total tocopherols	1.92 ^C^	2.09 ^C^	2.94 ^A^	2.65 ^B^	0.100	<0.001
β-Carotene	Nd	Nd	Nd	Nd	-	-
Retinol	0.77 ^C^	1.11 ^B^	2.12 ^A^	2.26 ^A^	0.150	<0.001

Nd, not detected. Different superscripts within the same row are significant or trending (A/B: *p* ≤ 0.01). SEM—standard error of the mean. Groups: RB, hybrid ryegrass cv. Bakus; FB, *Festulolium* cv. Becva; FF, *Festulolium* cv. Felopa; FP, *Festulolium* cv. Paulita. DM, dry matter; CA, crude ash; CP, crude protein; EE, ether extract.

**Table 5 animals-13-01817-t005:** Proximate chemical composition (%) and the content of β-carotene, tocopherols and retinol (mg/kg) in the *musculus longissimus lumborum* of rams.

Specification	Groups				SEM	*p*-Value
	RB	FB	FF	FP		
DM	27.01	26.64	26.36	25.93	0.242	0.153
CA	1.08	1.10	1.03	1.06	0.018	0.601
CP	21.84	21.51	20.84	21.75	0.152	0.073
EE	4.09	4.03	4.49	3.12	0.250	0.272
α-Tocopherol	2.09 ^B^	2.31 ^B^	3.53 ^A^	3.05 ^A^	0.150	<0.001
β-Tocopherol	0.09 ^B^	0.05 ^B^	0.19 ^A^	0.21 ^A^	0.018	<0.001
γ-Tocopherol	0.36 ^C^	0.27 ^D^	0.50 ^B^	0.71 ^A^	0.039	<0.001
δ-Tocopherol	0.22 ^C^	0.27 ^C^	0.82 ^A^	0.42 ^B^	0.056	<0.001
Total tocopherols	2.75 ^B^	2.90 ^B^	5.03 ^A^	4.39 ^A^	0.236	<0.001
β-Carotene	nd	nd	Nd	nd	-	-
Retinol	0.74	1.19	0.78	1.21	0.083	0.053

Nd-not detected. Different superscripts within the same row are significant or trending (A/B/C/D: *p* ≤ 0.01). SEM—standard error of the mean. Groups: RB, hybrid ryegrass cv. Bakus; FB, *Festulolium* cv. Becva; FF, *Festulolium* cv. Felopa; FP, *Festulolium* cv. Paulita. DM, dry matter; CA, crude ash; CP, crude protein; EE, ether extract.

**Table 6 animals-13-01817-t006:** Coefficients of correlation between tocopherols and β-carotene intake and the content of β-carotene, tocopherols and retinol in the *musculus gluteus maximus* of rams.

	α-Tocopherol	β-Tocopherol	γ-Tocopherol	δ-Tocopherol	Total Tocopherols	β-Carotene	Retinol
α-Tocopherol	1.00						
β-Tocopherol	−0.94 *	1.00					
γ-Tocopherol	0.87 *	−0.95 *	1.00				
δ-Tocopherol	0.90 *	−0.87 *	0.68 *	1.00			
Total tocopherols	0.99 *	−0.94 *	0.88 *	0.89 *	1.00		
β-Carotene	-	-	-	-	-	1.00	
Retinol	0.93 *	−0.75 *	0.71 *	0.75 *	0.92 *	0.91 *	1.00

* *p* < 0.05; r < 0.2 no correlation, 0.2–0.4 weak positive correlation, 0.4–0.7 moderate positive correlation, 0.7–0.9 fairly strong positive correlation, r > 0.9 very strong positive correlation. A minus sign indicates a negative correlation.

**Table 7 animals-13-01817-t007:** Coefficients of correlation between the tocopherols and β-carotene intake and the content of β-carotene, tocopherols and retinol in the *musculus longissimus lumborum* of rams.

	α-Tocopherol	β-Tocopherol	γ-Tocopherol	δ-Tocopherol	Total Tocopherols	β-Carotene	Retinol
α-Tocopherol	1.00						
β-Tocopherol	0.86 *	1.00					
γ-Tocopherol	0.67 *	0.94 *	1.00				
δ-Tocopherol	0.94 *	0.69 *	0.41 *	1.00			
Total tocopherols	0.99 *	0.91 *	0.73 *	0.92 *	1.00		
β-Carotene	-	-	-	-	-	1.00	
Retinol	−0.04	−0.01	0.23	−0.31	−0.06	0.92 *	1.00

* *p* < 0.05; r < 0.2 no correlation, 0.2–0.4 weak positive correlation, 0.4–0.7 moderate positive correlation, 0.7–0.9 fairly strong positive correlation, r > 0.9 very strong positive correlation. A minus sign indicates a negative correlation.

## Data Availability

The data presented in this study are available from the corresponding author.

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
