# Peer review of "The Effect of Festulolium Silage-Based Diets on the Content of Tocopherols, β-Carotene and Retinol in Meat from Young Rams"

_animals, 2023, doi:10.3390/ani13111817_

Round 1

Reviewer 1 Report

I found the paper to be very valuable. Results were accurately reported, and findings were discussed and compared with those in other published papers. A few modifications are needed for the manuscript to be improved further.

P1,L10 = Simple Summary

Change “Grasses of the genus Festulolium are intergeneric hybrids of fescue (Festuca L.) and ryegrass (Lolium L.) that combine the complementary traits of both genera, including high nutritional value and resistance to adverse weather conditions.” to be “A Festulolium grass is an intergeneric hybrid of fescue (Festuca L.) and ryegrass (Lolium L.) that combines the nutritional qualities and weather resistance of each.”.

P1,L24 = Abstract

It would be better if the abstract was rewritten, and the accomplishments should be statistically significant and clearly communicated.

P1,L37 = Introduction

Please try to use newer references. Please indicate the reasons for using this type of feed in rams feed.

You can use new references such as:

Ahmadzadeh-Gavahan, L., Hosseinkhani, A., Palangi, V. & Lackner, M. (2023). Supplementary feed additives can improve lamb performance in terms of birth weight, body size, and survival rate. Animals, 13(6).

Palangi, V., & Lackner, M. (2022). Management of Enteric Methane Emissions in Ruminants Using Feed Additives: A Review. Animals, 12(24), 3452.

P5,L194 = Results

Results explained adequately.

P8,L266 = Discussion

Discussion explained adequately. Please try to use newer references.

P10,L379 = Conclusions

Conclusions explained adequately.

P11,L409 = References

References are adequate. Please try to use newer references.

Regards

Author Response

Thank you for your comments and taking the time to read our manuscript. Your review will help us improve the quality of our work. 

P1,L10 = Simple Summary

Change “Grasses of the genus Festulolium are intergeneric hybrids of fescue (Festuca L.) and ryegrass (Lolium L.) that combine the complementary traits of both genera, including high nutritional value and resistance to adverse weather conditions.” to be “A Festulolium grass is an intergeneric hybrid of fescue (Festuca L.) and ryegrass (Lolium L.) that combines the nutritional qualities and weather resistance of each.”.

Authors’ response: The required change has been made (L 10-11).

P1,L24 = Abstract

It would be better if the abstract was rewritten, and the accomplishments should be statistically significant and clearly communicated.

Authors’ response: Abstract has been rewritten, focusing on statistically significant results (L 24-37).

P1,L37 = Introduction

Please try to use newer references. Please indicate the reasons for using this type of feed in rams feed. You can use new references such as:

Ahmadzadeh-Gavahan, L., Hosseinkhani, A., Palangi, V. & Lackner, M. (2023). Supplementary feed additives can improve lamb performance in terms of birth weight, body size, and survival rate. Animals, 13(6).

Palangi, V., & Lackner, M. (2022). Management of Enteric Methane Emissions in Ruminants Using Feed Additives: A Review. Animals, 12(24), 3452.

Authors’ response: The required content (reasons for using Festulolium in ram feed) has been added (L58-73). As suggested, we have replaced some references with more recent ones (No. 1, 6, 9, 10, 17, 18, 19, 20, 44, 45). Unfortunately, despite our efforts, we were not able to eliminate all older references. In some cases, this was due to the fact that research on the topic has not always been updated in recent years. Secondly, we believe that the scientific merit of older publications cited in our paper remains high.

P5,L194 = Results

Results explained adequately.

Authors’ response: Thank you for the positive feedback.

P8,L266 = Discussion

Discussion explained adequately. Please try to use newer references.

Authors’ response: We have replaced some of the older publications with more recent ones, but not all of them, for the reasons given in an earlier reply.

P10,L379 = Conclusions

Conclusions explained adequately.

Authors’ response: Thank you for the positive feedback.

P11,L409 = References

References are adequate. Please try to use newer references.

Authors’ response: References have been updated, and newer references cited in the text have been added to the reference list.

Reviewer 2 Report

The comments and suggestions are highlighted in the pdf archive.

Reviewer 3 Report

This study could be published if authors include data on intake and digestibilities, without it is very difficult to justify these results. I have found a lot of mistakes in the manuscript, especially sentence structure is too poor to read. I would suggest authors revise the manuscript carefully, add data and improve the manuscript and resubmit it. 

Below are my general comments

Line 12: extreme or adverse ‘…value and resistance to adverse weather conditions….’

Line 21: optimum results of what ? make sure, you are specific with your objective ‘Optimal results were obtained when silage was made from Festulolium cvs. Felopa and Paulita’

Line 24-25: please change the sentence ‘This study analyzed the chemical composition and the content of β-carotene, tocopherols and retinol in meat…..’ to ‘This study analyzed the chemical composition and β-carotene, tocopherols and retinol contents in meat…………….’

Line 29-30: P value is missing ‘The meat of rams fed Festulolium silage-based diets contained more tocopherols and retinol’

Line 30-31: what do you mean by ‘However, increased intake of tocopherols and β-carotene did not always lead to increased concentrations of tocopherols and retinol in meat’ it doesn’t show your treatment etc. please be specific with your results

Line 31-32: results doesn’t show any clarity, please rewrite it ‘Optimal results 31 were obtained when silage was made from Festulolium cvs. Felopa and Paulita’

Line 32-34: p value missing ‘Diets based on silage 32 made from different Festulolium cultivars did not affect the proximate chemical composition of 33 meat’

Resutls don’t have the materials and methods section. For example how silage was prepared, how many animals were slaughtered? Please include necessary details

Line 46: please use suitable words ‘adverse weather conditions’

Line 48: please rephrase it ‘…; they are also a source of palatable fodder for..’

Line 50-51: what do you mean by ‘Adaptation to environmental stresses is an important con sideration in the era of global climate change’ is this about plant or animals. Every sentence should be clear

Line 56-57: please correct the sentence structure ‘In most cases, it is cut and fed fresh as roughage or ensiled and fed as silage’

Line 57-58: what do you mean by ‘it’ in the sentence ‘It can also be offered to cattle and sheep as hay and haylage’. Reference is also missing

Line 58-59: Reference is missing ‘In Poland, lambs are usually fattened on hay or haylage supplemented with oat or barley grain’

Line 59: Replace ‘Roughage is’ with ‘Roughage are’

line 62: what do you mean by ‘They are’ please be specific with your terms

line 62-64: sentence is not clear ‘They are fat-soluble and share several common mechanisms underlying their metabolism and transfer to the offspring’ please rewrite it
line 66-67: please provide reference and also clarify the meaning ‘These compounds must be provided in the diet because they cannot be synthesized by the body’

line 67-68: rewrite the sentence ‘They are accumulated in the tissues and transferred to animal  products such as meat, improving its quality and nutritional value’ the new sentence could be ‘It has been reported that vitamin A and E accumulate in the tissues and transferred to animal  products such as meat, and known improve meat quality and nutritional value’

Line 68-71: meanings are not clear ‘The proximate chemical composition of Festulolium-based diets has been extensively researched, but very studies have investigated their content of bioactive compounds, including carotenoids and tocopherols’ please rewrite the sentence

Line 72: ‘in meat’ of ‘what’?

Line 101-102: please explain it ‘During the study, the animals were cared for in accordance with the Act of 15 January 2015 on the Protection of Animals Used for Scientific or Educational Purposes’

Line 103-105: were animal randomly selected and than divided into experimental treatments, if yes, than its not clear in sentence, if no, please explain how did you made the groups ‘At the beginning of the experiment, the 103 average body weight of animals in each group was as follows: RB - 33 ± 6 kg, FB - 32 ± 5 104 kg, FF - 31 ± 6 kg, FP - 32 ± 4 kg’

Line 105: what do you mean by ‘The lambs were kept in individual boxes..’ please give dimension and use suitable words for boxes

Line 106-107: please use suitable word for leftover ‘Feed intake and leftovers were recorded throughout the study’ and explain how feed intake was determined

Line 107-108: what does DLG reflect for ? please explain and make sentences clear ‘The diets for growing sheep were formulated in accordance with the DLG  system [24]’

Line 110-111: replace ‘The diets contained also ground….’ With ‘The diets also contained ground……………….’

Line 135: what does ‘Analytical feed samples’ represent

Line 297-299: important data of intake is missing which can correlate this study ‘The inclusion of Festulolium-based silage in ram diets increased the concentrations of both total tocopherols and α-tocopherol in meat, despite their lower intake’

Author Response

Thank you for your comments and taking the time to read our manuscript. Your review will help us improve the quality of our work.

This study could be published if authors include data on intake and digestibilities, without it is very difficult to justify these results. I have found a lot of mistakes in the manuscript, especially sentence structure is too poor to read. I would suggest authors revise the manuscript carefully, add data and improve the manuscript and resubmit it.

Authors’ response: Data on feed and nutrient intake have been added (L209-220). In response to your concerns over the quality of the English language used in our manuscript, we would like to clarify that since the authors are not native speakers of English, the manuscript has been translated by a professional translator who has extensive experience in editing scientific manuscripts. It has also been spell-checked and grammar-checked by a native English speaker prior to re-submission (please see the attached language editing certificate). Unfortunately, we disagree with the comment that vitamin digestibility data should be included in our study. In order to obtain reliable results on vitamin digestibility, it would be necessary to examine vitamin bioavailability - for example, changes in the chemical composition of gastric/ rumen feed samples in transfused animals, enzyme activity, vitamin levels in the blood. Such measurements/methodologies are related to the fact that digestibility studies are not applicable to vitamins, or to other biologically active compounds. Such data are certainly very interesting, but they were not the subject of our study.

Line 12: extreme or adverse ‘…value and resistance to adverse weather conditions….’

Authors’ response: The sentence has been changed (L12)

Line 21: optimum results of what ? make sure, you are specific with your objective ‘Optimal results were obtained when silage was made from Festulolium cvs. Felopa and Paulita’

Authors’ response: The sentence has been changed (L20-22).

Line 24-25: please change the sentence ‘This study analyzed the chemical composition and the content of β-carotene, tocopherols and retinol in meat…..’ to ‘This study analyzed the chemical composition and β-carotene, tocopherols and retinol contents in meat…………….’

Authors’ response: The sentence has been changed (L24-26).

Line 29-30: P value is missing ‘The meat of rams fed Festulolium silage-based diets contained more tocopherols and retinol’

Authors’ response: P values have been added (L29-31).

Line 30-31: what do you mean by ‘However, increased intake of tocopherols and β-carotene did not always lead to increased concentrations of tocopherols and retinol in meat’ it doesn’t show your treatment etc. please be specific with your results

Authors’ response: The sentence has been removed.

Line 31-32: results doesn’t show any clarity, please rewrite it ‘Optimal results 31 were obtained when silage was made from Festulolium cvs. Felopa and Paulita’

Authors’ response: We have changed the sentence (L34-35).

Line 32-34: p value missing ‘Diets based on silage 32 made from different Festulolium cultivars did not affect the proximate chemical composition of 33 meat’

Authors’ response: P value has been added (L37).

Results don’t have the materials and methods section. For example how silage was prepared, how many animals were slaughtered? Please include necessary details

Authors’ response: The manuscript includes a Materials and Methods section (L84-207). It is divided into subsections (Experimental silages; Animals, Experimental design and diets; Feed sampling and analyses, Meat sampling and analyses, Statistical analysis), in which all the required information is included. In addition, the Materials and Methods section has been completed as required by other Reviewers. Information about silage production is included in the “Experimental silages” subsection and we have added the information on the number of slaughtered animals (L 123).

Line 46: please use suitable words ‘adverse weather conditions’

Authors’ response: The relevant correction has been made (L49).

Line 48: please rephrase it ‘…; they are also a source of palatable fodder for..’

Authors’ response: The relevant correction has been made (L51).

Line 50-51: what do you mean by ‘Adaptation to environmental stresses is an important con sideration in the era of global climate change’ is this about plant or animals. Every sentence should be clear

Authors’ response: The sentence has been changed (L53-54).

Line 56-57: please correct the sentence structure ‘In most cases, it is cut and fed fresh as roughage or ensiled and fed as silage’

Authors’ response: The sentence has been changed (L59-60).

Line 57-58: what do you mean by ‘it’ in the sentence ‘It can also be offered to cattle and sheep as hay and haylage’. Reference is also missing

Authors’ response: The sentence has been changed. The reference has been added (L59-60).

Line 58-59: Reference is missing ‘In Poland, lambs are usually fattened on hay or haylage supplemented with oat or barley grain’

Authors’ response: The relevant reference has been added (L62).

Line 59: Replace ‘Roughage is’ with ‘Roughage are’

Authors’ response: “Roughage is” is a grammatically correct form; “roughage” can be used with a singular verb; it is both a countable and an uncountable noun; when roughage is used as a countable noun, the plural form is roughages.

line 62: what do you mean by ‘They are’ please be specific with your terms

Authors’ response: The sentence has been changed (L63-65).

line 62-64: sentence is not clear ‘They are fat-soluble and share several common mechanisms underlying their metabolism and transfer to the offspring’ please rewrite it

Authors’ response: The sentence has been rewritten. (L65-66).

line 66-67: please provide reference and also clarify the meaning ‘These compounds must be provided in the diet because they cannot be synthesized by the body’

Authors’ response: The sentence has been changed and the relevant reference has been added (L68-69).

line 67-68: rewrite the sentence ‘They are accumulated in the tissues and transferred to animal  products such as meat, improving its quality and nutritional value’ the new sentence could be ‘It has been reported that vitamin A and E accumulate in the tissues and transferred to animal  products such as meat, and known improve meat quality and nutritional value’

Authors’ response: The sentence has been changed and we have added more information as suggested by another Reviewer (L69-72)

Line 68-71: meanings are not clear ‘The proximate chemical composition of Festulolium-based diets has been extensively researched, but very studies have investigated their content of bioactive compounds, including carotenoids and tocopherols’ please rewrite the sentence

Authors’ response: The sentence has been rewritten (L74-77).

Line 72: ‘in meat’ of ‘what’?

Authors’ response: The sentence has been changed (L77).

Line 101-102: please explain it ‘During the study, the animals were cared for in accordance with the Act of 15 January 2015 on the Protection of Animals Used for Scientific or Educational Purposes’

Authors’ response: The Act of January 15, 2015 on the Protection of Animals Used for Scientific or Educational Purposes contains a list of guidelines for the maintenance conditions of experimental animals to ensure high animal welfare. We have included the relevant information in the manuscript to assure the reader that the ethical standards were observed in accordance with the current Polish law. Below are examples of other papers where a similar declaration was included:

  • Purwin, C.; Starczewski, M.; Borsuk, M.; Nogalski, Z.; Opyd, P.M.; Mazur-Kuśnirek, M.; Białobrzewski, I. The Quality, Intake, and Digestibility of Virginia Fanpetals (Sidahermaphrodita Rusby) Silage Produced under Different Technologies and Its Effect on the Performance of Young Cattle. Animals 2021, 11, 2270.
  • Czurgiel S, Antoszkiewicz Z, Mazur-Kuśnirek M, Bogdaszewski M. The Effect of the Inclusion of Different Concentrates in Feed Rations on the Contents of Tocopherols, β-Carotene and Retinol in the Livers and Longissimus dorsi Muscles of Farm-Raised Fallow Deer (Dama dama). Animals. 2022; 12(23):3311.
  • Purwin, C., Borsuk-Stanulewicz, M., Nogalski, Z., Baranowska, M., Zygmuntowicz, A., and Michalski, J. P.: Digestibility and palatability of Virginia fanpetals (Sida hermaphrodita) silage in sheep, Arch. Anim. Breed., 65, 89–96, https://doi.org/10.5194/aab-65-89-2022, 2022.

Line 103-105: were animal randomly selected and than divided into experimental treatments, if yes, than its not clear in sentence, if no, please explain how did you made the groups ‘At the beginning of the experiment, the 103 average body weight of animals in each group was as follows: RB - 33 ± 6 kg, FB - 32 ± 5 104 kg, FF - 31 ± 6 kg, FP - 32 ± 4 kg’

Authors’ response: The missing information has been added (L116).

Line 105: what do you mean by ‘The lambs were kept in individual boxes..’ please give dimension and use suitable words for boxes

Authors’ response: The sentence has been changed and more information has been added (L109-112).

Line 106-107: please use suitable word for leftover ‘Feed intake and leftovers were recorded throughout the study’ and explain how feed intake was determine

Authors’ response: The word has been changed and more information has been added (L112-114).

Line 107-108: what does DLG reflect for ? please explain and make sentences clear ‘The diets for growing sheep were formulated in accordance with the DLG  system [24]’

Authors’ response: DLG-tables of feed value and feeding standards for ruminants - is one of the systems for balancing energy and nutrients for ruminants (including sheep). DLG was developed in 1997 by the German Agricultural Society and published in 1999 in Poland. DLG contains principles for estimating feed value and standardizing feed, and it is one of the best systems that is most applicable under Polish conditions, not only due to the geographical proximity of the two countries, and similar climatic and soil conditions, but also because similar livestock diets are fed in both countries. The DLG system is based on the Dutch estimation method proposed by VAN ES (1978), which has been modified over the past 20 years, and whose reliability has been confirmed in numerous studies. The sentence has been changed (L114-115).

Line 110-111: replace ‘The diets contained also ground….’ With ‘The diets also contained ground……………….’

Authors’ response: The sentence has been changed (L119).

Line 135: what does ‘Analytical feed samples’ represent

Authors’ response: The sentence has been changed (L143-144).

Line 297-299: important data of intake is missing which can correlate this study ‘The inclusion of Festulolium-based silage in ram diets increased the concentrations of both total tocopherols and α-tocopherol in meat, despite their lower intake’

Authors’ response: Data on nutrient intake have been added (L209-220).

Reviewer 4 Report

 It is well written articel. I have only a few comments below for the authors consideration:

1- can you add/discuss if this grass is avaialble worldwide since it will be useful for the internatioanl reader also.

2- is there any comparative study available internatioanlly?

3- Based on the given results, can you add/dicuss how the results can be implemented and adapted by the farmers/industry?

4- can you add more figure to make more visible for the readers?

5- may i suggest to discuss the possible implications of this stiudy both locally and internationally.

Author Response

Thank you for your comments and taking the time to read our manuscript. Your review will help us improve the quality of our work.

1- can you add/discuss if this grass is available worldwide since it will be useful for the international reader also.

Authors’ response: This information is included in the Introduction section (L 42-44).

2- is there any comparative study available internationally?

Authors’ response: To the best of the authors' knowledge, no similar studies can be found in publicly available scientific databases. A domestic scientific study (Pieta, M., & Patkowski, K. (2007) Jakość mięsa jagnięcego w zależności od żywienia matek. Żywienie Człowieka i Metabolizm3(34), 1309-1315.) has been found, however it investigated the sensory quality of meat from lambs whose mothers were fed rations containing Festulolium haylage. Also, we have recently studied the same chemical components (tocopherols, β-carotene and retinol) in the meat of farmed fallow deer, but the animals were fed different rations (Czurgiel, S., et al. 2022. The Effect of the Inclusion of Different Concentrates in Feed Rations on the Contents of Tocopherols, β-Carotene and Retinol in the Livers and Longissimus dorsi Muscles of Farm-Raised Fallow Deer (Dama dama L.). Animals12(23), 3311). We have not come across other studies investigating the direct effect of Festulolium forage on the concentrations of tocopherols, retinol and β-carotene in sheep meat.

3- Based on the given results, can you add/discuss how the results can be implemented and adapted by the farmers/industry?

Authors’ response: The required information has been added (L297-304; L374-376; L405-410).

4- can you add more figure to make more visible for the readers?

Authors’ response: The figure has been added (L207).

5- may i suggest to discuss the possible implications of this study both locally and internationally

Authors’ response: The required information has been added (L405-410).

Reviewer 5 Report

The objective of this study was to evaluate the influence of Festulolium silage-based diets on the content of basic nutrients, β-carotene, tocopherols and retinol in the musculus longissimus lumborum and musculus gluteus maximus of rams.

The study is very simple, reporting only β-carotene, tocopherols and retinol content in two muscle. They did not provide detail information on the feed intake and other characteristics of meat, etc.

Authors could have analyzed β-carotene, tocopherols and retinol in different tissues such as liver, heart, kidney, fatty tissues to make the study informative and interesting.

Table 5 and 6 are not needed. What information does it provide if the intake is not provided. The content of a chemical does not correlate unless total intake is not provided. How could authors explain negative correlations obtained in the study in Table 5, but not in Table 6?

Author Response

Thank you for your comments and taking the time to read our manuscript. Your review will help us improve the quality of our work.

The objective of this study was to evaluate the influence of Festulolium silage-based diets on the content of basic nutrients, β-carotene, tocopherols and retinol in the musculus longissimus lumborum and musculus gluteus maximus of rams.

Authors’ response: We agree with the Reviewer.

The study is very simple, reporting only β-carotene, tocopherols and retinol content in two muscle. They did not provide detail information on the feed intake and other characteristics of meat, etc.

Authors’ response: We agree with this comment. We have added more information about feed and nutrient intake (L209-220. Other data, including meat quality traits, would undoubtedly be interesting. However, the authors were looking for answers to simple research questions regarding the concentrations of selected nutrients in meat in relation to nutrition. Other questions were not the subject of this study, but perhaps the research will be continued in the future.

Authors could have analyzed β-carotene, tocopherols and retinol in different tissues such as liver, heart, kidney, fatty tissues to make the study informative and interesting.

Authors’ response: Thank you for this suggestion, we agree that such a study would have high scientific merit and would be very interesting to both researchers and readers. In the current study, we looked for answers to simple research questions regarding the concentrations of selected ingredients in meat only. However, the above issues could be addressed in future research.

Table 5 and 6 are not needed. What information does it provide if the intake is not provided. The content of a chemical does not correlate unless total intake is not provided. How could authors explain negative correlations obtained in the study in Table 5, but not in Table 6?

Authors’ response: We have added information about feed and nutrient intake (L209-220). Tables 5 and 6 contain information about two different muscles for which the results may differ due to, among others, their different activity and chemical composition. In the present study, we focused on determining the concentrations of selected components in meat depending on sheep's diet. In order to find out more, a follow-up study should be extended.

Round 2

Reviewer 3 Report

Dear Authors, 

overall manuscript is improved. However, some results in abstract section are missing especially newly added data. Furthermore, P values are missing in the abstract and I would request to improve it. Moreover, correlate your new results with previously described results in discussion section.

Regards 

Author Response

Dear Authors, 

overall manuscript is improved. However, some results in abstract section are missing especially newly added data. Furthermore, P values are missing in the abstract and I would request to improve it. Moreover, correlate your new results with previously described results in discussion section.

Regards

Authors’ response: Thank you for your time to reviewing our work. Your suggestions are very valuable. We have tried to include all statistically significant results in the abstract section, but we are limited to 200 words as required by the journal editors. Because we obtained many statistically significant differences in all five tables (intake of all forms of tocopherols and total tocopherols, concentration of all forms of tocopherols, total tocopherols and retinol in MGM, concentration of all forms of tocopherols and total tocopherols in MLL, almost all coefficients r in both muscles), we had to selected the most important (in our opinion) results. We have selected the most important and representative results from each table, and we had to present them in the general way. Our aim was also to present the information in a way that is clear to the reader. Given the above, we have added a general description of the significant results obtained from Table 3 (newly added data about feed intake) (L 29-30), two sentences describing the significant results from Tables 4 and 5 (L 30-35) and a sentence about the most important results from Tables 6 and 7 (L 35-37). We have added the p values to each result (L 30, L 31, L 33) and r coefficients (L 36, L 37). The new results about feed and nutrients intake are described and correlated with the nutrients concentration in meat and selected correlations in the discussion section (L 359-361; L 363-377; L 389-394; L 397-404; L 427-440).

Reviewer 5 Report

Authors have improved the manuscript. But still some improvement is needed.

1. How were animals fed the diets? Were feed amounts recorded by individual animals? In Table 3, SEM for DM intake is too low. With this SEM values, the treatments would be significantly different. I have doubt of putting SEM values.

2. Table 6 and 7: Authors should present correlations between intake versus content in meat instead of content in feed versus content in meat.  Content in feeds does not imply unless intake is correlated.

3. The negative correlations values in the tables are not clear. How could a negative correlation be present unless they are interfering each other in the absorption. I do not think so that they really interfere in the absorption. Please explain with references.

Table 4: Please explain why DM content tended to be different in gluteus muscle by treatments but not in longissimum.

Author Response

Authors have improved the manuscript. But still some improvement is needed.

Authors’ response: Thank you for your time to reviewing our work. Your suggestions are very valuable.

1. How were animals fed the diets? Were feed amounts recorded by individual animals? In Table 3, SEM for DM intake is too low. With this SEM values, the treatments would be significantly different. I have doubt of putting SEM values.

Authors’ response: During the experiment, each animal was kept in individual cages (one animal in one cage), with an automatic drinker and a trough in each cage. Through the automatic drinker the animals had access to water at all times (water flowed out of the drinker when the animal pressed the drinker). Each animal was given feed in a trough placed in the cage. Before putting the feed into each trough, each feed was weighed. The uneaten feed from each trough was weighed each day before new feed was given. We distributed the feed daily, twice a day (in the morning at 7 a.m. and in the afternoon at 5 p.m.), in sufficient quantity to leave some uneaten feed. We have not included the above information in the text of the article because our experience shows that such detailed methodological descriptions are not usually included in the text of papers. Practice shows that only the most important information is included. Information on animal maintenance and nutrition is described in lines 120-125. In some previously published papers, for example Purwin C, Opyd PM, Baranowska M, Borsuk-Stanulewicz M. The Effect of Diets Containing High-Moisture Corn or Triticale Grain on Animal Performance and the Fatty Acid Composition of Lamb Muscles. Animals. 2022; 12(22):3130. https://doi.org/10.3390/ani12223130 the description of the information discussed is similar to the description in our paper. In addition, we have added the information “one animal in each cage” in L 120 and word "each" in lines 122 and 123 to make the sentences very clear. We checked our statistical analysis, and found no error in the SEM values. Previous analyses using the same methods (one-way analysis, Duncan's test) showed no statistically significant differences at lower SEM values. An example is the work above, where the SEM for feed intake was 0.031 and the results were not statistically significant. The standard error denotes a measure of the statistical precision of the estimate and determines how accurate the sample mean is to the true population mean. The lower the SEM, the closer the data is to that of the whole population.

  1. Table 6 and 7: Authors should present correlations between intake versus content in meat instead of content in feed versus content in meat.  Content in feeds does not imply unless intake is correlated.

Authors’ response: As you suggested, we did the correlation between nutrients intake and nutrients content in the meat. As you can see in the tables, values did not change significantly - probably changes between the intakes data and their concentration in the feed were not sufficient to produce different correlation results. It should be noted that the concentration of the compounds in the meat remained the same.

3. The negative correlations values in the tables are not clear. How could a negative correlation be present unless they are interfering each other in the absorption. I do not think so that they really interfere in the absorption. Please explain with references.

Authors’ response: The negative correlation may be due to, among other things, competition of nutrients during absorption. Tocopherols and carotenoids (especially b-carotene) are well described examples of competition in the literature. This phenomenon is discussed among others by Yang et al. (2002) in Effect of vitamin E supplementation on a-tocopherol and b-carotene concentrations in tissues from pasture- and grain-fed cattle. As Yang writes "a-tocopherol, like other lipids and fat-soluble vitamins, is dependent upon micellar formation for transport across intestinal membranes (Hollander, 1981) where it is incorporated into lipoproteins and secreted into the intestinal lymph (Bjørneboe, Bjørneboe, & Drevon, 1990) for distribution to other tissues. High concentrations of other components compete for these sites in the micelle and in the absorption and transport process. Knight and colleagues (Knight and Death, 1999; Knight, Death, Muir, Ridland, & Wyeth, 1996) observed that oral vitamin A significantly reduced the concentration of plasma carotenoids in grazing cattle and Pellett et al. (1994) described an interaction between b-carotene and a-tocopherol in chickens." It is worth mentioning that issues concerning the absorption of biologically active ingredients (including carotenoids, tocopherols and vitamins) are not fully explained in the literature and have been better studied in monogastric animals than in ruminants. It should be emphasised, however, that issues from the bioavailability of the tested components were not the focus of the present study.

Table 4: Please explain why DM content tended to be different in gluteus muscle by treatments but not in longissimum.

Authors’ response: In the present study, the MGM of all groups contained less protein and more fat (no significant results) than the MLL, resulting in a lower dry matter content in the MGM. To clarify the reasons for the differences in chemical composition between the two muscles, it would be necessary to extend the present study, as factors other than nutrition affecting the differences in chemical composition between MGM and MLL should be taken into account. However, it is natural that the chemical composition is not the same in all carcass muscles. Not all of the issues involved have been described in the literature, but it is known that the chemical composition of meat varies according to the type of muscle and its location in the carcass, which is due to differences in muscle conformation and its lifetime functions. The muscles in the carcass vary in structure as they contain different proportions of muscle fibers and connective, adipose, vascular, and nervous tissues. Muscle fibres, intramuscular connective tissue, and intramuscular fat play key roles in the determination of meat quality, including nutritional value derived from chemical composition. (Joo, S. T., Kim, G. D., Hwang, Y. H., & Ryu, Y. C. (2013). Control of fresh meat quality through manipulation of muscle fiber characteristics. Meat science95(4), 828-836.; Listrat, A., Lebret, B., Louveau, I., Astruc, T., Bonnet, M., Lefaucheur, L., ... & Bugeon, J. (2016). How muscle structure and composition influence meat and flesh quality. The Scientific World Journal2016.).